# Atomically precise graphene etch stops for three dimensional integrated systems from two dimensional material heterostructures

Jangyup Son[1], Junyoung Kwon[2], SunPhil Kim[1], Yinchuan Lv[3], Jaehyung Yu[1], Jong-Young Lee[2], Huije Ryu[2], Kenji Watanabe [4], Takashi Taniguchi[4], Rita Garrido-Menacho[3,5], Nadya Mason[3,5], Elif Ertekin[1,5], Pinshane Y. Huang[5,6], Gwan-Hyoung Lee [2] & Arend M. van der Zande[1,5]

Atomically precise fabrication methods are critical for the development of next-generation technologies. For example, in nanoelectronics based on van der Waals heterostructures, where two-dimensional materials are stacked to form devices with nanometer thicknesses, a major challenge is patterning with atomic precision and individually addressing each molecular layer. Here we demonstrate an atomically thin graphene etch stop for patterning van der Waals heterostructures through the selective etch of two-dimensional materials with xenon difluoride gas. Graphene etch stops enable one-step patterning of sophisticated devices from heterostructures by accessing buried layers and forming one-dimensional contacts. Graphene transistors with fluorinated graphene contacts show a room temperature mobility of $40,000\,cm^2\,V^{-1}\,s^{-1}$ at carrier density of $4 \times 10^{12}\,cm^{-2}$ and contact resistivity of $80\,\Omega\cdot\mu m$. We demonstrate the versatility of graphene etch stops with three-dimensionally integrated nanoelectronics with multiple active layers and nanoelectromechanical devices with performance comparable to the state-of-the-art.

[1] Department of Mechanical Science and Engineering, University of Illinois at Urbana-Champaign, 1206 W Green Street, Urbana, IL 61801, USA. [2] Department of Materials Science and Engineering, Yonsei University, 50 Yonsei-ro, Seodaemun-gu, Seoul 03722, Korea. [3] Department of Physics, University of Illinois at Urbana-Champaign, 1110 W Green Street, Urbana, IL 61801, USA. [4] National Institute for Materials Science, 1-1 Namiki, Tsukuba, Ibaraki 305-0044, Japan. [5] Frederick Seitz Materials Research Laboratory, University of Illinois at Urbana-Champaign, 104 S Goodwin Avenue MC-230, Urbana, IL 61801, USA. [6] Department of Materials Science and Engineering, University of Illinois at Urbana-Champaign, 1304 W Green Street, Urbana, IL 61801, USA. These authors contributed equally: Jangyup Son, Junyoung Kwon. Correspondence and requests for materials should be addressed to G.-H.L. (email: gwanlee@yonsei.ac.kr) or to A.M.V.D.Z. (email: arendv@illinois.edu)

As next-generation technologies of electronic, photonic, and mechanical devices approach the atomic scale, it is important to develop atomically precise fabrication methods. Among them, etch stops, critical for the vertical integration of nanoelectronic and nanomechanical devices, are created by layering materials with drastically different etch properties and embedded into a structure, allowing for patterning feature sizes, accessing buried layers, or undercutting to create suspended structures. Accordingly, improved fabrication techniques are especially needed in nanoelectronics based on van der Waals (vdW) heterostructures, where two-dimensional (2D) materials are stacked to form electronic devices with nanometer thicknesses[1–5]. Many of the applications for 2D material heterostructure devices demand out-of-plane integration, contacting multiple active layers, and creating interconnects between the different layers[1–5]. Examples include 2D material-based integrated circuitry like NAND gates[6] or ring oscillator[7]; devices based on interlayer tunneling like light-emitting diodes (LEDs)[8] or tunnel transistors[9]; and nanoelectromechanical systems (NEMS) like resonators based on atomic membranes[10].

In the current state of the art, the relative ease of assembly of vdW heterostructures, which occurs through the sequential pick-up and stamped release of individual atomic layers, contrasts starkly with the difficulty of patterning and electrically addressing each layer in a heterostructure device. As a result, the majority of studies in this field are performed on monolayers or on heterostructures where each layer has been carefully offset so they are accessible for electrical contacts through direct deposition of metal on top or on the exposed edges of individual 2D layers[11–14]. The current state-of-the-art method is to use edge contacts where heterostructures are etched through to expose the edges of buried layers of graphene encapsulated in insulating hexagonal boron nitride (hBN), and then metals are evaporated onto the edge to make one-dimensional (1D) contacts[13]. This method has led to a dramatic improvement in the mobility and quality of electronic devices because it allows contact to electronic layers that are fully encapsulated and thus have a minimum of disorder[13,15]. However, edge contacts still require careful offsetting of active layer because the etching is not selective so all vertically aligned layers in the heterostructure are exposed simultaneously. A method that combines the superior device behavior of the edge contacts but that simultaneously allows ready patterning of 2D heterostructures from large area continuous sheets and individually addressing of each layer are critical for translating many of the recent demonstrations of this class of devices into scalable technologies.

Here we show methods to fabricate nanostructures and access buried interfaces with the precision of a single atomic layer by using graphene as impermeable etch masks and etch stops[16,17]. These techniques, which we call GES (graphene etch stops), represent a straightforward method to selectively expose and contact embedded graphene layers within 2D heterostructures. This concept takes advantage of the high chemical selectivity of $XeF_2$, a vapor phase, strong fluorinating agent commonly used as an isotropic etchant for silicon in the microelectromechanical systems (MEMS) industry[18]. Several 2D materials including hBN and transition metal dichalcogenides, are quickly etched when exposed to $XeF_2$[16,19,20]. In contrast, graphene reacts with $XeF_2$ to form fluorographene (FG)[21–24], a wide band-gap semiconducting monolayer[21,22] with composition $C_4F$, in the case that only one side is exposed[22]. There have been several demonstrations that take advantage of this selectivity to use graphene as an etch mask for shaping $MoS_2$[16], as a mask to etch underlying silicon[25–27], and to create a sacrificial release layer to suspend graphene membranes on silicon on insulator[17,22]. Our innovation has been to apply this etch selectivity to access buried graphene layers

embedded within the heterostructures and as masks for patterning the underlying layers. Surprisingly, the embedded contacts, which is composed of FG–metal contacts, lead to room temperature carrier mobilities of 40,000 cm$^2$ V$^{-1}$ s$^{-1}$ at carrier density $n = 4.0 \times 10^{12}$ cm$^{-2}$ and behave as 1D contacts with low contact resistivity of 80 Ω μm, approaching theoretical limits[11,28]. This capability enables simple and scalable methods to vertically integrate 2D devices through contacting multiple active layers, interlayer vias, and suspended nanostructures, yet maintains the state-of-the-art performance of fully encapsulated 2D devices.

## Results

**Selective etching by graphene etch stop**. Figure 1a illustrates the use of GES to pattern a heterostructure of 2D materials, and Fig. 1b, c are optical images of the same heterostructure before and after exposure to $XeF_2$. The heterostructure is fabricated by stacking individual materials using established polymer-free, aligned transfer techniques[11,29]. Specifically, the heterostructure is composed of two monolayer graphene flakes, set in a cross-alignment and embedded between hBN layers, then placed on top of a Si/SiO$_2$ (285 nm) substrate. The color corresponds with the thickness of the heterostructure. In Fig. 1b, two graphene layers are not visible because their contrast is completely overwhelmed by the much thicker hBN, but the inset Raman map confirms their positions. As seen in Fig. 1c, after exposure to $XeF_2$ (3 Torr for 30 s at room temperature), the exposed hBN is completely etched, while the graphene layers and the hBN underneath them remain. In Supplementary Note 1, Supplementary Figure 1, and Supplementary Table 1, we show that the same process can be applied to many other 2D material heterostructures as well, including hBN, $MoS_2$, $WSe_2$, and black phosphorus (BP).

To examine the selectivity and resolution limits of GES, we obtained cross-sectional images of the etched heterostructures with a scanning transmission electron microscope (STEM), as shown in Fig. 1d, e. Importantly, the hBN layers under the FG show no etching, indicating that they are protected from the $XeF_2$. Moreover, Fig. 1e shows that the buried layer of graphene (G1) is unaffected by the etch process and there are atomically sharp and clean interfaces between stacked graphene and hBN layers (see also Raman data in Supplementary Figure 2). These results demonstrate that FG maintains the impermeable nature of graphene[30,31] through the chemical modification process[16,32]. In Fig. 1d, at the edge of the etch mask, the underlying hBN has a sub-nanometer slope. These images demonstrate that GES is a self-arresting etch process that enables atomic precision out of plane and nanometer-scale feature sizes without requiring precise control in timing or conditions.

**Characteristics of fluorinated graphene etch stops**. The self-arresting nature of GES means that it is scalable as well as being atomically precise. Fig. 2a demonstrates this scalability by applying GES to a large area heterostructure array. We patterned large area graphene as etch masks for patterning large area $WS_2$, both grown by chemical vapor deposition (CVD). First, large area continuous graphene was patterned into lines with lithography and oxygen plasma. Two sets of the patterned lines were then sequentially transferred onto large area continuous monolayer $WS_2$ grown by CVD. The second transfer is set perpendicular to the first to form a cross-hatch pattern. After exposure to $XeF_2$, $WS_2$ uncovered by graphene was completely etched, while the region covered by graphene remained under FG (process flow in Supplementary Figure 3 and photoluminescence maps in Supplementary Figure 4). This process of combining prepatterned graphene masks with the selective etch can be repeated to scalably achieve arbitrarily complex heterostructures of layered FG,

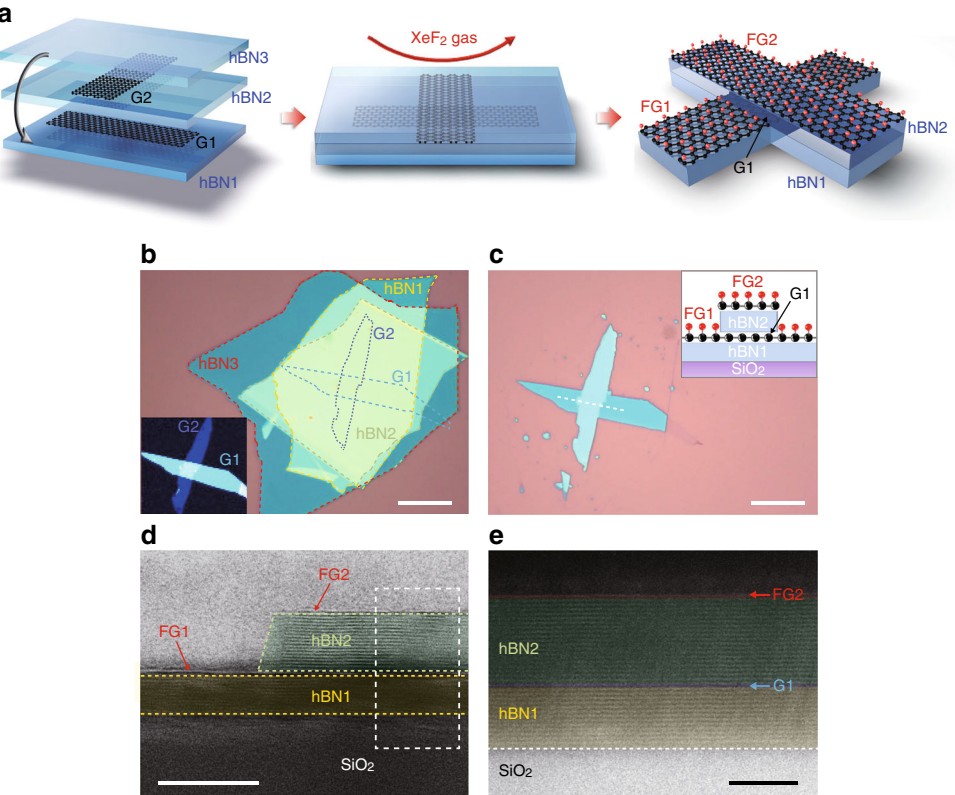

**Fig. 1** Selective etching of a vdW heterostructure with XeF$_2$ gas. **a** Schematic of the XeF$_2$ etching process for a vdW heterostructure of stacked hBN and graphene layers. **b**, **c** Optical micrographs of a corresponding heterostructure fabricated from stacked exfoliated flakes before and after exposure to XeF$_2$. The scale bar is 10 μm. Before etching, the heterostructure is composed (from bottom to top) of silicon oxide substrate, 5 nm hBN, 1 L graphene, 8 nm hBN, 1 L graphene, and 10 nm hBN. The inset in **b** shows a Raman map of the 2D graphene peak, indicating the positions of the two graphene layers (G1 and G2). **c** Optical micrograph of the sample after XeF$_2$ etching, with an inset illustration indicating the cross-sectional structure in the region indicated. The changes in color between **b** and **c** represent changes in film thickness as determined by thin-film interferometry. The substrate is brown, while the thinnest hBN is blue and increasing thickness, and the changes in color represent changes in the hBN thickness from dark blue (thinnest) to light blue to green to yellow (thickest). **d** False-color cross-sectional bright-field STEM image of the etched heterostructure. The scale bar is 10 nm. The hBN layers (hBN1 and hBN2) covered with graphene masks (FG1 and FG2) were protected from XeF$_2$ etching. **e** Annular dark-field STEM image taken from the white-dashed area of (**c**) shows atomically sharp and clear heterointerfaces. The scale bar is 5 nm

graphene, and other materials on the wafer-scale patterns, which could not be realized with conventional patterning and etching techniques.

Before examining the application of GES to 2D heterostructure devices, we confirm the structure and electrical properties of FG. Figure 2b shows the Raman spectra of graphene on hBN under increasing exposure to XeF$_2$. Initially, only the G and 2D peak are visible, indicating clean graphene with no defects, as well as one additional peak from the underlying hBN. After exposure to XeF$_2$, the D peak appeared and the 2D peak was suppressed. Both phenomena are a result of the breaking of hexagonal symmetry within the graphene lattice due to the formation of $sp^3$ bonds by bonding of fluorine atoms onto the graphene surface[22]. Supplementary Figures 5 and 6 show additional structural analyses of X-ray photoelectron spectroscopy (XPS) and TEM and demonstrate that the fluorination condition results in only a $sp^3$-type lattice transition without formation of voids, consistent with the observation that FG acts as an impermeable barrier[16]. Figure 2c shows the electrical transport through a prefabricated graphene transistor on hBN (device shown in the inset and additional details in Supplementary Figure 7). Before fluorination, the graphene shows mS conductance and gate dependence typical of the linear dispersion in graphene band structure. The device conductance drastically decreased as a function of fluorination

time. After 10 s, the device conductance decreased by a factor of 10, while after 30 s, the graphene became insulating with resistance exceeding 60 GΩ. As shown in Supplementary Figure 8, when graphene on hBN is functionalized for longer periods (720 s), it maintained its structure and high resistance for over 2 months in ambient conditions.

**Electrical properties of fluorinated graphene contacts**. In the rest of the paper, we will explore the application of GES to fabricating electronic and mechanical devices from 2D heterostructures. Two persistent challenges in nanoscale device research are how to minimize the impact of environment on limiting the potentially outstanding electronic mobility of nanomaterials and how to engineer low resistance contacts to nanomaterials. Previous studies have shown that achieving the theoretical limits of performance in graphene devices requires graphene to never come into contact with solvents or polymers and charged impurity scattering to be suppressed by fully encapsulating the samples in hBN[13]. However, doing this brings a challenge of how to electrically contact the encapsulated graphene layers. In Fig. 3a, we demonstrate the application of GES to electrically contact a buried graphene layer encapsulated in hBN. Using e-beam lithography, electrodes were patterned on top of hBN/G/hBN heterostructure. Then the structure was exposed to XeF$_2$ before

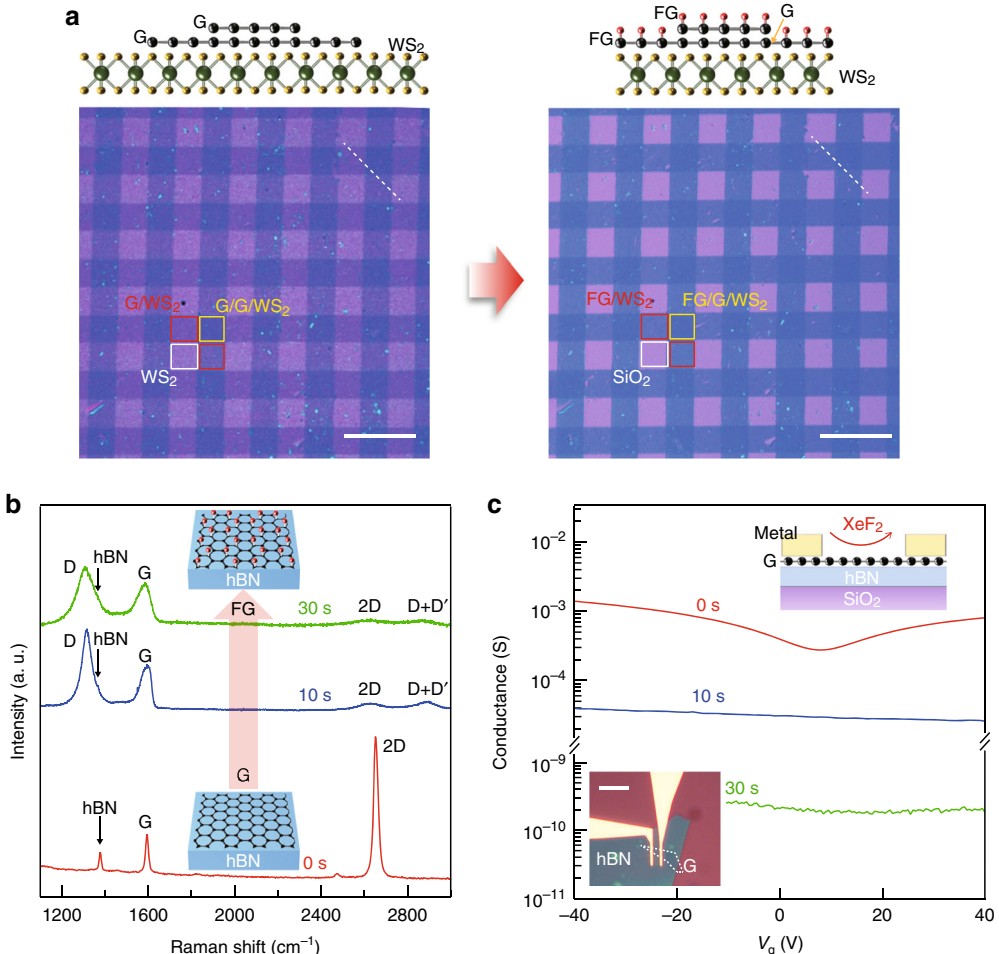

**Fig. 2** Raman and conductance measurements of fluorinated graphene. **a** Optical images (bottom) of CVD $WS_2$ sheet covered with prepatterned CVD GES before and after $XeF_2$ etching (the scale bar is 50 μm). Schematic illustrations (top) show cross-section of graphene-covered $WS_2$ along white dashed lines for each step. After etching, uncovered $WS_2$ is etched away, meanwhile the covered $WS_2$ remains unchanged under protection of fluorinated GES. **b** Raman spectra of graphene on hBN under increasing exposure to $XeF_2$. As $XeF_2$ exposure time increases, the graphene D peak becomes prominent while the 2D peak is damped, indicating the formation of $sp^3$-type defects. These results are consistent with other studies of graphene fluorination. **c** Electrical conductance of graphene on hBN fluorinated by $XeF_2$ treatment (the scale bar in the inset is 10 μm). After a 30 s $XeF_2$ exposure, FG becomes fully insulating

metallization. As it is well known, the $XeF_2$ does not attack the polymer. However, within the patterned regions, the top hBN is etched away, locally exposing and fluorinating the buried graphene layer. Electrodes were then deposited through the same polymer mask directly on to the fluorographene regions (1 nm Cr, 30 nm Pd, and 40 nm Au; see Supplementary Figure 9 and Methods for details). Figure 3a shows a cross-sectional high-resolution transmission electron microscope (HR-TEM) image of the FG electrical contact. The lithographic pattern is transferred into the hBN, and the evaporated metal is deposited only on the exposed FG, while the graphene channel under the hBN is never exposed.

Figure 3b shows the field-effect characteristics of a graphene Hall bar device encapsulated by hBN with FG contacts. The mobility was calculated by the Drude model, $\mu = \sigma/ne$ where $\mu$, $n$, $e$, and $\sigma$ are the carrier mobility, carrier density, electron charge, and sheet conductivity, respectively. At high carrier concentration of $n = 4.0 \times 10^{12}$ $cm^{-2}$, the sheet resistance was 45 Ω per square at room temperature, corresponding to a carrier mobility of 40,000 $cm^2$ $V^{-1}$ $s^{-1}$, close to the theoretical limit[28]. As shown in the inset of Fig. 3b, the mobility drastically increases with

decreasing carrier concentrations, as expected from the acoustic-phonon-limited model[13,28]. On a Hall bar device measured at low temperature $T = 1.7$ K (Supplementary Figure 10), the low carrier concentration mobility increased to 460,000 $cm^2$ $V^{-1}$ $s^{-1}$. This mobility corresponds with a mean free path of 4.6 μm, similar to the channel width of the device so the mobility is likely limited by device dimensions rather than material properties. The device conductance ($I_{ds} - V_{ds}$) is linear and displayed no hysteresis (Supplementary Figure 11). The contact resistance of the FG to the buried graphene channel was quantified by performing transfer length measurements on the device shown in the inset of Fig. 3c. Figure 3c shows the resistance vs. channel length at different charge concentrations. The contact resistances are extracted from the extrapolated zero-length intercepts to get 21 Ω for holes and 49 Ω for electrons for a 4 μm wide channel. Figure 3d shows the contact resistance vs. carrier concentration from 2.4 K to room temperature. The contact resistance vs. temperature is shown in the Fig. 3d inset. The resistivity is not significantly affected by temperature and can reach a value of 80 Ω·μm at $n = 4.0 \times 10^{12}$ $cm^{-2}$, which means absence of the potential barrier at the contact. This is distinct from the graphene

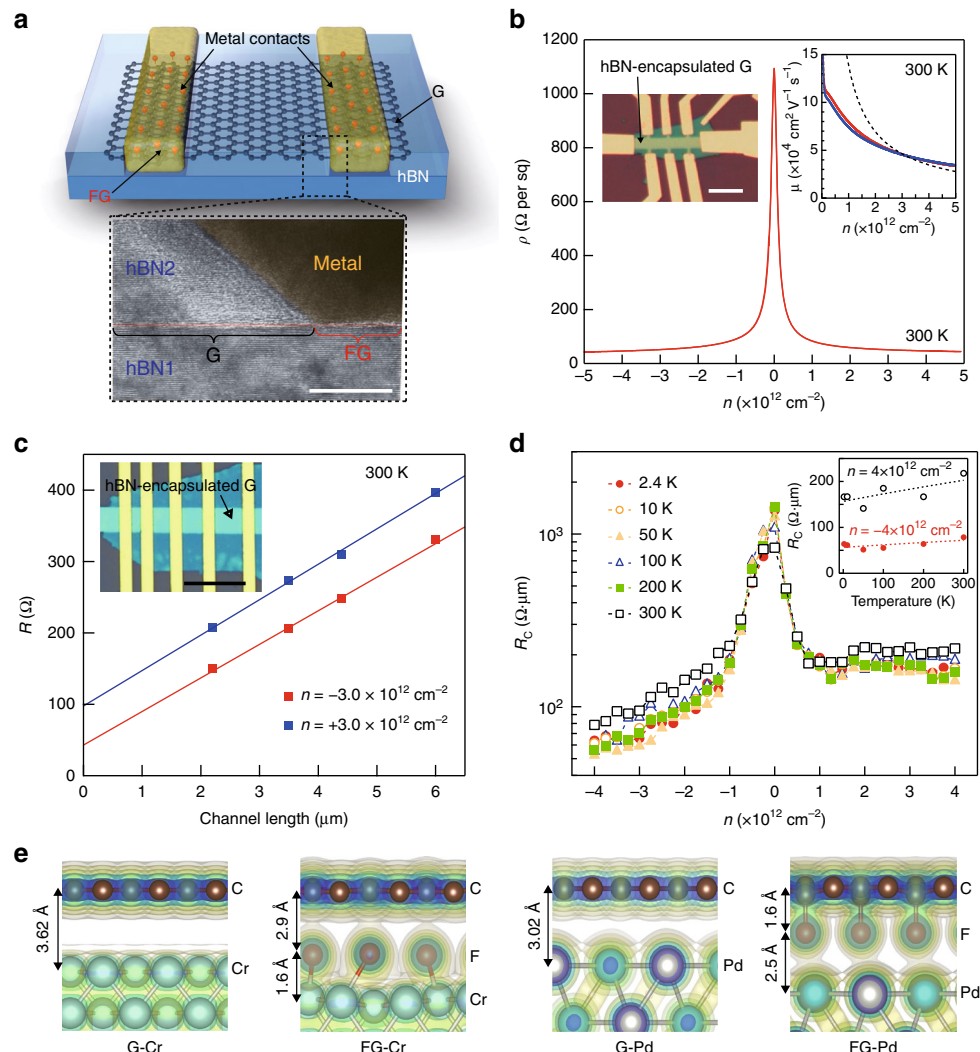

**Fig. 3** Electrical performance of hBN-encapsulated graphene device with FG via contacts. **a** Schematic of a hBN-encapsulated graphene device with FG via contacts and false-color cross-sectional HRTEM image of the FG via contact region (the scale bar is 5 nm). **b** Four-probe resistivity as a function of carrier density at room temperature. The right inset shows a Hall bar graphene device used for carrier mobility measurement (the scale bar is 5 μm). Right inset shows the electron (red) and hole (blue) mobilities extrapolated by applying the Drude model to the measured conductivity ($\sigma = ne\mu$, where $\sigma$, $n$, $e$, and $\mu$ are the sheet conductivity, carrier density, electron charge, and carrier mobility, respectively). Black dashed line shows the predicted intrinsic phonon limited mobility of graphene at room temperature[28]. **c** Plot of total resistances of the graphene TLM device as a function of channel length, at fixed electron and hole carrier densities. The inset shows optical micrograph of the TLM device (the scale bar is 10 μm). **d** Contact resistances of the device as a function of carrier density and temperature. The inset shows contact resistances as a function of temperature and indicates no significant change in contact resistance. **e** Isosurfaces of the total charge densities at the interfaces between G–Cr, FG–Cr, G–Pd, and FG–Pd, calculated with DFT. The shortened atomic distances of C–F–Cr and C–F–Pd at the interfaces of FG–metal contacts lead to small contact resistance. This results because orbital overlap through a bridge of F facilitates charge transfer from metals to FG

devices with surface-contacted metal electrodes, which show temperature dependence of contact resistance due to the potential barrier formed at the contact[13]. In addition, all the devices display robust chemical and electrical stability. The devices showed no significant change after 1 month, when stored in air (Supplementary Figure 12). Taken together, the outstanding mobility, low contact resistance, and stability from encapsulation make these devices comparable to the state of the art across all metrics (Supplementary Figure 13 and Supplementary Table 2 contain a comparison), while simultaneously being much easier to fabricate due to the self-arresting mechanism of GES. However, these low contact resistances are surprising, especially given the high in-plane resistance of FG measured in Fig. 2b.

To explain the low contact resistance, density functional theory was applied to simulate the interlayer distance and charge

distribution at a FG and metal heterointerface. Figure 3e shows the equilibrated structure and local density of states in four different interfaces: either graphene or FG and either Cr or Pd metal (see Supplementary Figure 14 and Methods for simulation details). The 1 nm-thick Cr adhesion layer forms islands, not a continuous film (Supplementary Figure 15), so both Cr and Pd will make direct contact to the FG surface. From a Landauer framework, the factors governing contact resistance are related to the carrier transmission probability $T$ and the number $M$ of conduction modes available[12]. Transport must occur both from the metal to the graphene under the metal and from the graphene under the metal to the channel region, which have different transmission probabilities. Additionally, the number of graphene conduction modes under the metal is reduced in some cases due to charge transfer doping by the metal. The contact resistance can

be improved by achieving a smaller effective metal/graphene coupling length to increase $T$ or by finding metals that lead to high metal-induced doping concentrations to increase $M$. From the isosurfaces, without fluorine the G–Cr interface shows weak vdW bonding[33], and the small orbital overlap leads to low $T$ due to the presence of a tunneling barrier and a large effective coupling distance[34]. The G–Pd interface shows somewhat more orbital hybridization between the graphene $p_z$ and metal d states, consistent with prior results[33]. Interestingly, when fluorinated the degree of hybridization increases for both metals, particularly for FG–Pd, indicating a reduced coupling length that renders the charge transfer more ballistic and lowering the contact resistance[34]. Additionally, the fluorine hybridization opens a band gap in graphene, reducing its work function, and inducing a large effective n-type charge transfer doping. The combination of the strong coupling and the enhanced charge transfer doping suggest that the contact resistance is limited by the FG–graphene interface rather than the metal–FG interface, resulting in 1D edge contacts rather than 2D surface contacts. To test this hypothesis, a variable width channel device was fabricated. The contact resistance was linearly proportional to the reciprocal of channel width (Supplementary Figure 16) as expected in 1D contacts. These simulations and measurements show that the dominant contact resistance in the FG-buried devices is at the 1D graphene–FG interface. Similar results have been seen in electrically contacting the sides of etched heterostructures where only the 1D edge of graphene is exposed[13]. Unlike in 2D surface contacts, this result indicates that the contact resistance of the metallized FG is independent of the contact length and there is no intrinsic lower limit of size. Hence, it should be possible to scale down the size of embedded contacts and vias to nanoscale dimensions without impacting functionality.

**Fabrication of three-dimensional (3D) integrated systems from 2D materials**. In addition to offering a simpler fabrication process and state-of-the-art device properties, the selective etch stop also enables capabilities that cannot be easily or scalably realized using other techniques. For example, interlayer vias and independently contacting multiple active layers in vertically aligned heterostructures are critical to integrated circuits like NAND gates[6], where logic operations are computed by coupling the gates and channels of several transistors in series, and graphene-based multilayered printed circuit boards (PCBs). Similarly, many device applications of 2D heterostructures that rely out-of-plane transport, like vertical PN junctions[35], tunnel junctions[9], or light emitters based on 2D materials[8,36] require the same ability to contact vertically aligned layers separately.

Figure 4 outlines proof-of-concept demonstrations of using GES to fabricate interlayer vias and vertically integrate multiple active layers. First, the GES is a self-arresting process that allows access to multiple buried layers set at different depths within a single etch step. This allows the creation of interlayer vias, which are critical components for integrated circuits, where wiring and devices can operate on more than one plane. Figure 4a is an optical image of a multilayer graphene–hBN heterostructure with interlayer vias fabricated in a single lithography and etch step identical to the one used in Fig. 3 on a single layer. The heterostructure is formed by sequentially stacking three graphene layers, each separated by few-layer hBN. Each graphene layer operates as a separate transistor, and the out-of-plane vias are formed by exposing two layers within a single opening before metallization. Figure 4b is the corresponding transfer curves within each graphene device. In this particular geometry, the layers are offset, so they may all be controlled with the global backgate. The inset of Fig. 4b is the interlayer transport current

through the vias vs. interlayer bias, which shows a linear dependence. For example, the total resistance measured from the labeled electrodes, B1–M1, is $1\,k\Omega$ at $V_g = 0\,V$, equivalent to the in-plane channel resistance of the graphene. The contact resistance is negligibly small compared with the corresponding channel resistances, demonstrating that GES enables efficient, simple, and selective contacts to vertically offset layers to create low resistance interlayer vias, which could be integrated to the complicated devices such as multilayer PCBs or light emitter based on 2D materials.

The second demonstration takes advantage of the combined high in-plane resistance of FG with the low contact resistance when the FG is metallized. Through sequential patterning and etching steps, GES allows independent contact of multiple active layers that interact to generate device functionality. This allows the creation of 3D integrated circuitry from 2D materials, where, for example, vertically offset encapsulated 2D layers act as both the gate and channel in a transistor, which has been difficult to realize with conventional patterning techniques or 1D edge contacting[13] (Supplementary Figure 17). Figure 4c shows the optical image and schematic illustration of a hBN-encapsulated graphene transistor channel with graphene backgate (i.e. two coupled active layers). In the heterostructure, the two graphene sheets are separated by hBN. Key to this demonstration is that the top graphene layer is larger than the bottom layer and fully covers it. In order to access the buried bottom layer, the selective etch process is repeated twice with an oxygen plasma shaping step in between, then lastly depositing electrodes contacting all layers (see Supplementary Figure 18 for the full fabrication process). The high in-plane and vertical resistance of both the top FG and the dielectric hBN allow access to the buried bottom layer without shorting the two layers together. Figure 4d is the transfer curve of the resulting embedded all 2D material field effect transistors, showing that the top graphene channel (G2) can be effectively modulated by the bottom graphene gate (G1). Like the single layer demonstration in Fig. 3, all 2D layers are encapsulated, protecting them from extrinsic disorder and resulting in high carrier mobilities. The two demonstrations above prove that the selective etch process enables interlayer vias and 3D integration of multiple active device layers made entirely of 2D materials, both capabilities that are critical to the development of integrated circuitry from 2D materials.

As a final demonstration of a different kind of vertical integration, a common application of selective etches in MEMS is to suspend mechanically responsive structures. In studies on this, it was shown that graphene can be used as an etch mask for underlying silicon to generate suspended FG membranes[17,22]. These atomic membranes behave as tensioned mechanical resonators[37], useful as low mass chemical sensors or tunable radio frequency filters or oscillators. In Fig. 4e, we show that the same concept can be applied to fabricate suspended graphene NEMS from 2D heterostructures. Figure 4e is an angled scanning electron microscopic (SEM) image of a suspended few-layer graphene membrane clamped by graphite supports. This membrane was fabricated by first creating a heterostructure of narrow few-layer graphene ribbon on 70-nm-thick black phosphorus, with 100 nm thick graphite at either end. The black BP etches far more quickly than other 2D materials when exposed to $XeF_2$ (Supplementary Figure 1 and Supplementary Table 1), allowing it to act as a sacrificial release layer that undercuts the graphene (see fabrication process in Supplementary Figure 19). The resulting FG membrane was fully suspended without wrinkles and contamination in a dry vapor phase process. Figure 4f is the mechanical resonance of the membrane measured using modulated laser optomechanical actuation and dynamic reflection contrast detection[10,37] (Supplementary Figures 20 and

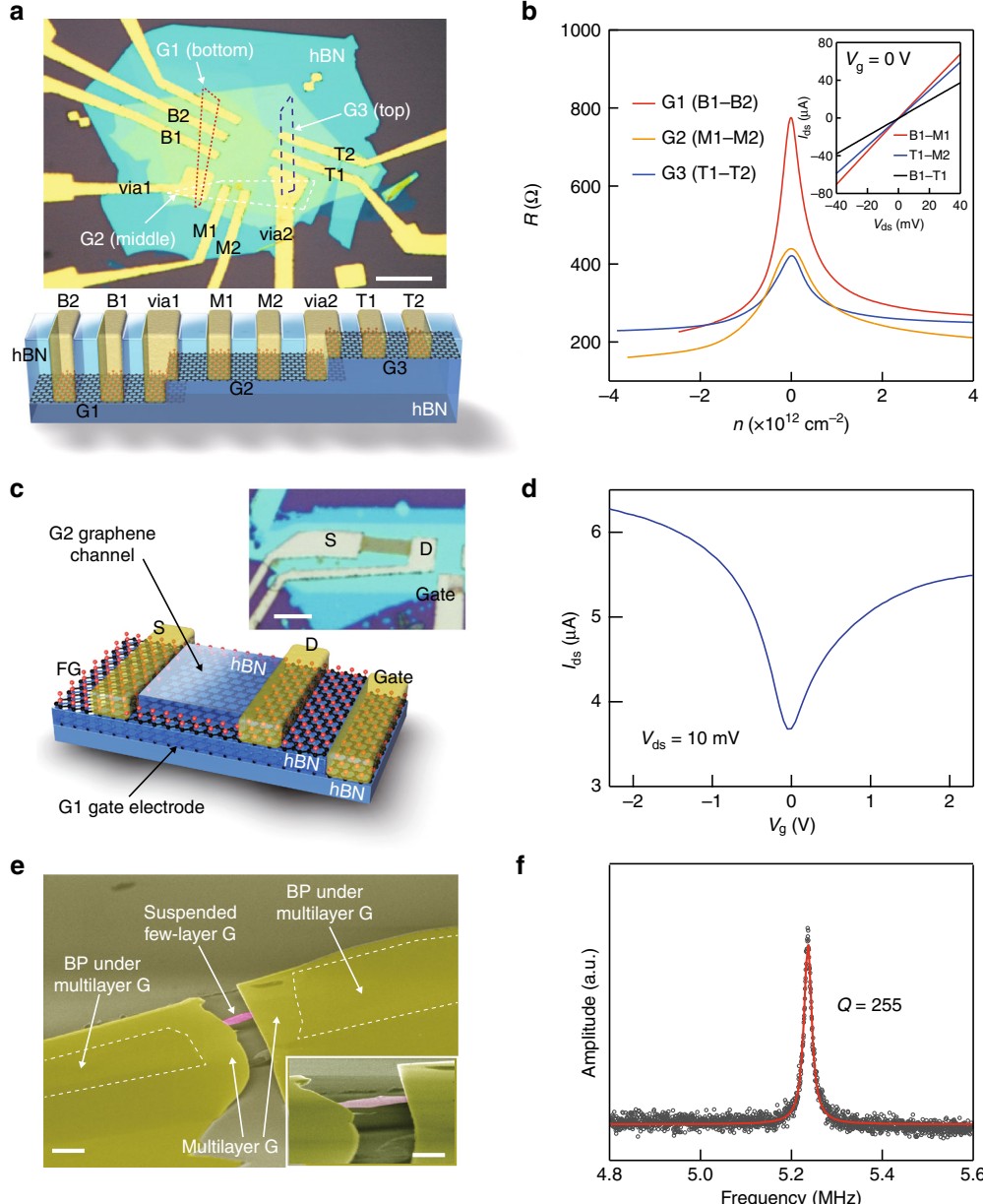

**Fig. 4** Fabrication of 3D integrated devices from 2D materials and suspended graphene mechanical resonators with a dry and one-step etching process. **a** Optical microscopic image and schematic illustration of the multi-stacked graphene devices connected with via contacts (the scale bar is 10 μm). Source and drain contacts were denoted as B1 and B2 for bottom graphene device (G1), M1 and M2 for middle graphene device (G2), and T1 and T2 for top graphene device (G3). All the graphene devices are connected with two via contacts (via1 and via2). **b** Plots of resistance vs. carrier density of multi-stacked three graphene devices in **a**. The inset shows $I_{ds} - V_{ds}$ curves obtained from two graphene devices connected with via1 or via2. Linear curves and small resistance indicate that these three graphene devices embedded in hBN are electrically connected with low resistance via contacts. **c** Optical microscopic image and schematic illustration of the hBN-encapsulated graphene device with graphene backgate (the scale bar is 5 μm). **d** $I_{ds} - V_g$ curve of the graphene device (G2) in **c**. Gate voltage was applied with bottom graphene (G1). **e** False-color scanning electron micrograph of the suspended graphene membrane (the scale bar is 2 μm). The inset is a magnified image of the suspended membrane (the scale bar is 1 μm). **f** Plot of normalized amplitude vs. frequency of the few-layer graphene resonator. Black circles and red solid line are the optomechanical response and Lorentzian fit, respectively

21 and Methods). The graphene membrane has a resonant frequency of $f_1 = 5.24$ MHz and quality factor of 255 at room temperature, comparable to state-of-the-art graphene resonators with similar dimensions produced via a wet process or mechanical exfoliation over trenches[10,38]. Just as XeF$_2$ has found wide applicability in MEMS or NEMS industry as a selective silicon etch, using GES to produce graphene-based resonators has a great potential since the whole process is liquid free and clean.

Moreover, this route can produce suspended graphene with much higher aspect ratios and gap depths than with conventional transfer or wet etching techniques.

## Discussion

Taken together, the demonstrations in Fig. 4 show that GES proposed in this work enable advanced fabrication of

3D-integrated electronic and mechanical devices based on 2D materials. Figure 3 shows that the structures will maintain the high mobility and low contact resistances that are currently the state of the art. Figure 1 shows that nanometer scale in-plane features and devices should be possible. Most of the demonstrations above use exfoliated materials, but as shown in Fig. 2a, the scalability of this technique means that all demonstrations will also work on arrays of devices patterned from continuous, large area heterostructures. The self-arresting nature of this process means that precision is not needed to achieve uniformity of devices, a huge benefit in atomically precise electronics. All of these components suggest that the selective etch process is a major capability necessary for the realization of atomically precise, all-2D nanoelectronics as a viable technology, in applications where vertical integration is critical, like integrated circuit logic components (e.g., NAND gates), devices operating through out-of-plane transport (e.g., 2D material tunnel junctions and LEDs), and in 2D nanoelectromechanical systems. Finally, many other materials such as transition metals, silicon, and MBE grown III–V materials are also etched by XeF$_2$, so GES may find broad application for the fabrication of atomically precise devices beyond just 2D materials.

## Methods

**Fabrication of vdW heterostructures**. To fabricate the heterostructures, we use a 2D material pick-up technique with similar established methods[13,29]. Before creating the heterostructure, it is necessary to fabricate a sacrificial transfer substrate. First, a 0.5-mm-thick polydimethylsiloxane (PDMS) droplet is deposited on a microscope glass slide, then cured overnight at 60 °C. At the same time, poly (bisphenol A carbonate, Sigma Aldrich) (PC) dissolved in chloroform is deposited onto a microscope slide glass. The chloroform is allowed to evaporate in air at room temperature, then the remaining PC film is manually peeled off by hand. The peeled-off PC film is placed onto the PDMS, then the entire structure is baked at 170 °C for 15 min to form conformal contact between PC film and PDMS. The resulting transfer substrate is then fixed to a micromanipulator. In parallel, all 2D flakes used for the vdW heterostructures were separately exfoliated onto the SiO$_2$ (285 nm)/Si substrates with the scotch tape method. The thickness or layer number of each material is separately confirmed using a combination of Raman spectroscopy, atomic force microscopy, and optical microscopy. For the first pick-up, it is necessary to start with an extra thick layer of hBN (~20 nm). The PC/PDMS stamp is placed onto the target hBN flake at 70 °C. To increase adhesion strength between PC and hBN, the temperature is then raised to 130 °C. Then PC/PDMS stamp is gradually lifted up during cooling to 70 °C. This process is then repeated to pick up other 2D flakes subsequently at 90 °C. Each 2D piece must be smaller than the top layer of hBN. After stacking, the stacked heterostructure was transferred onto a clean SiO$_2$/Si substrate by releasing the PC film from the PDMS at a higher temperature above 190 °C. Lastly, the PC film was removed by rinsing the sample in chloroform.

**Xenon difluoride etching**. The XeF$_2$ etcher (Xactix etching system) was used for the selective etching of 2D materials in pulse mode with $P_{XeF2} = 3$ Torr at room temperature. The pulse time for etching, i.e., exposure time was set according to the thickness of top layer of hBN, between 30 s, to 2 min. However, it should be noted that the exposure time is not proportional to etch rate because etching stops at the graphene layer.

**Device fabrication**. The e-beam lithography (EBL, TESCAN) was performed to generate patterns to selectively etch the vdW heterostructures. The exposure to the XeF$_2$ gas did not affect the ability to remove the poly (methyl methacrylate) (PMMA) used as an e-beam resist using normal solvents. For fabrication of the devices in Fig. 3, the vdW heterostructure was etched by first patterning the PMMA on top of the heterostructure, then exposing the entire structure to XeF$_2$. The top layers of the heterostructure were etched away exposing the contact area of the embedded graphene, which fluorinated during etching (Supplementary Figure 9). Then metals of Cr/Pd/Au (1 nm/30 nm/40 nm) were deposited using e-beam evaporator (Temescal six pocket e-beam evaporation systems). Finally, lift-off process was performed simply by soaking the samples in acetone.

**Sample preparation for TEM**. In the TEM images of Supplementary Figures 6 and 15, graphene was grown by CVD following standard recipes described in a previous paper[39]. The CVD graphene was then transferred onto a TEM grid. PMMA was spin-coated on the as-grown CVD graphene on a copper foil, followed by etching of copper in ammonium persulfate solution. After rinsing in multiple baths of deionized (DI) water, the graphene/PMMA film floating on DI water was scooped

with the TEM grid. PMMA film was removed by dipping it in acetone. The cross-section TEM specimens in Fig. 1d, e in the main text were prepared using FEI Helios 600i Dualbeam focused ion beam (FIB), using standard lift-out procedures with a final milling step of 2 kV to reduce surface damage. For Fig. 3a, cross-section TEM sample of the encapsulated graphene device was prepared with FIB (JIB-4601F, JEOL).

**High-resolution TEM**. STEM images in Fig. 1d, e were acquired with a 200 kV aberration-corrected JEOL 2200FS STEM. HR-TEM images in Supplementary Figures 6 and 14 were acquired on a Cs-corrected TEM (JEM-ARM200F, JEOL). The acceleration voltage was fixed at 80 kV to minimize damage of graphene by electron beam irradiation.

**Scanning electron microscopy**. The SEM images in Fig. 4e of the heterostructure stack on an SiO$_2$ substrate were acquired on a Hitachi S-4700 field-emission gun SEM with 2 kV accelerating voltage. The sample is tilted by 45° with respect to the beam direction. False coloring was added after data acquisition.

**Raman spectroscopy**. Raman measurements in Fig. 2b and Supplementary Figure 8 were acquired on a Renishaw using a 633 nm laser and an 1800 mm$^{-1}$ grating. To minimize damage of graphene by irradiation of the laser, a power of < 5 mW was used with an acquisition of 60 s.

**XPS analysis**. XPS measurement in Supplementary Figure 5 was acquired using a K-alpha XPS system (Thermo VG, UK). For this measurement, graphene grown by CVD was used. To prevent peak shift by charging effect of the substrate, the CVD graphene was transferred onto Au-coated SiO$_2$ substrate. We utilized monochromated Al as X-ray sources (Al Kα line: 1486.5 eV) and X-ray power of 12 kV and 3 mA. All measurements were carried out in vacuum ($P < 5 \times 10^{-9}$ mbar).

**Electrical measurements**. For the temperature-dependent electrical measurements in Fig. 3, the devices were placed on a commercial chip carrier with 32 leads and electrically contacted using aluminum wires with a wedge-wire bonder. Then the devices were loaded into cryostat, with a base temperature of 1 K. Conventional two-point and four-point lock-in measurements were performed using an SR830. For measurement of via and graphene gated devices in Fig. 4, measurements were performed in air at room temperature with a semiconductor parameter analyzer (Keithley 4200).

**Resonator measurements**. Two lasers of different wavelengths were focused on the center of the graphene membrane and used to actuate and detect the mechanical resonance. To actuate the membrane, a 623 nm diode laser was modulated electrically. The reflected light of a second 520 nm laser was monitored through the Si-based avalanche photodetector. The modulation frequency was tuned and monitored using a spectrum analyzer to find the resonance frequency. The measurements were performed in an optical cryostat at < 5 μTorr to reduce damping of the membrane.

**Simulation of the FG–metal interface**. The atomic-scale structure and charge distribution of the interface between metal and FG shown in Fig. 3e and Supplementary Figure 14 was simulated using density functional theory[40,41] implemented in VASP[42] in conjunction with projected augmented wave[43]. The generalized gradient approximation of Perdew–Burke–Ernzerhof[44] was applied to describe the exchange-correlation functional. An energy cutoff of 350 eV was chosen for the plane wave basis and achieves convergence of the total energy of both Pd and Cr and graphene sheet to within 0.01 eV of the total energy. Geometry optimization allowed relaxation until the forces on each atom were < 0.1 eV/Å. A 64-atom supercell of graphene and 7 layers of Pd (111) (or Cr (111)) were used to describe the metal electrode, which is sufficient to recover bulk-like properties in the interior of the slab. Within GGA-PBE, the graphene lattice constant was 2.47 Å, and in our simulations, the Pd metal electrodes are under 3% compressive strain due to lattice mismatch with the graphene membrane. In each supercell, 20 Å of vacuum is included to avoid interaction between adjacent images in the z-direction. A $4 \times 8 \times 1$ mesh was used to sample the system k-space. Furthermore, graphene or fluorinated graphene was introduced on both sides of the metal to maintain symmetry in the supercells and to avoid spurious introduction of electric fields and dipole moments across the supercell.

## Data availability

The authors declare that all data supporting the findings of this study are available within the paper and its supplementary information files.

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

## Acknowledgements

This research was primarily supported by the National Science Foundation MRSEC program under NSF Award Number DMR-1720633. Work at Yonsei was funded by Samsung Research Funding & Incubation Center of Samsung Electronics under Project Number SRFC-MA1502-12. Y.L. and P.Y.H. were supported by the Air Force Office of Scientific Research under award number FA9550-7-1-0213; R.G.M acknowledges funding by DOE Basic Energy Sciences under DESC0012649. This work was carried out in part in the Fredrick-Seitz Material Research Laboratory Central Facilities and the Micro and Nano Technology Laboratory at UIUC. K.W. and T.T. acknowledge support from the Elemental Strategy Initiative conducted by the MEXT, Japan and JSPS KAKENHI Grant Numbers JP15K21722.

## Author contributions

J.S., J.K., G.H.L., and A.M.v.d.Z. conceived and designed the study. J.S and J.K. fabricated samples and carried out experiments under the guidance of G.H.L. and A.M.v.d.Z. and with the help of the other authors. S.P.K. performed optomechanical measurement. Y.L. and P.Y.H. executed the STEM and SEM experiments. J.Y. and E.E. carried out simulation. J.Y.L. and H.R. assisted in the preparation of heterostructure samples. R.G.M. and N.M. carried out the low-temperature electrical measurements. K.W. and T.T. prepared high-quality hBN. All authors contributed to the discussion of this work. J.S., J.K., P.Y.H., G.H.L., and A.M.v.d.Z. wrote the manuscript.
