## [Peer Review File · Nature Communications]

Reviewers' comments:

Reviewer #1 (Remarks to the Author):

The manuscript by Son et al. outlines a technical process through which graphene can serve as an etch mask for lithographically fabricating 2D material devices. XeF₂ gas is known to etch materials such as h-BN, TMDs, and phosphorene, while only fluorinating graphene to leave its carbon backbone intact such that underlayers are protected from etching. Several devices are fabricated and electrically tested to show how the process proceeds under different geometrical considerations. Electrically tested graphene devices show high mobility after processing.

Overall the paper is well written and the experiments appear to be thoughtfully executed. The central point of graphene serving as an etch mask to XeF₂ is in and of itself not a new finding, though this work extends the examples and complexity through which graphene can be used to form device structures, in particular focusing on multi-layer, metal-patterned structures. As such, the introduction must make this point clearer with a few sentence discussion/referencing of how graphene has been used in the past specifically as an etch mask with XeF₂. This work has the potential to further motivate researchers to implement graphene/XeF₂ etching techniques.

For example: (i) In Reference 5, Figure 4 (and Figure S4) they use graphene with O₂ plasma etched holes to XeF₂ etch a MoS₂ underlayer, which is analogous to the experiment shown in this manuscript in Figure 2; (ii) In Reference 22, Figure 2 they use graphene as an etch mask and SOI as a sacrificial underlayer, as well as patterning the fluorine itself on graphene using photoresist; (iii) In Nano Lett. 12, 4212 (2012), Figure 2 they use reduced graphene oxide and in ACS Nano 11, 4745 (2017), Figure 1 they use CVD graphene with lithographically etched holes to undercut a SOI sacrificial underlayer to form arrays of mechanical drum resonators, which is analogous to the experiment shown in this manuscript in Figure 4. These last references also study how chemistry impacts graphene's properties in mechanical resonators, which is relevant for graphene mechanical structures formed via XeF₂ etching.

Another point not explicitly discussed relates to the stability of graphene fluorinated by XeF₂ [Reference 8]. It would be helpful to the reader to note that the fluorination is somewhat unstable and will change over time in ambient, as well as to subsequent processing in lithography-based solvents or vacuum. If the reader expects to have a 'locked-in' graphene fluorination level after XeF₂ exposure and processing, that will likely not happen.

Reviewer #2 (Remarks to the Author):

The authors present a technique to selectively etch and pattern hexagonal boron nitride (hBN) and tungsten disulfide (WS₂) by using xenon difluoride (XeF₂) gas and a graphene etch mask (GEM). They claim that graphene is fluorinated but not etched by the XeF₂, and acts as an impermeable barrier to the XeF₂ gas. Therefore, the XeF₂ etches the hBN or WS₂ that is not protected or masked by the GEM. In any case, they use the GEM to deterministically etch hBN and WS₂. The authors argue that the GEM technique addresses a need to address individual layers of a 2D crystal heterostructure.

To show that the graphene becomes fluorinated, the authors perform Raman and XPS, and they also fabricate a G/hBN FET. Raman and XPS clearly show fluorination directly or indirectly, and the conduction of the FET falls sharply with increased XeF₂ exposure. To show that G isn't etched, they perform TEM and also show optical images of the devices. From both, it is clear that G is not etched. They also show TEM cross-sections of etched hBN and WS₂, and demonstrate nanometer-scale control and a very small undercut. The authors also claim that the fluorination of graphene is continuous, which they attempt to support with TEM; I'm not convinced by this claim, and wonder

what happens to the amorphous carbon (PMMA residue) on the graphene. Does it become fluorinated and thus protect the graphene below it? Perhaps the authors could soften the claims of uniformity, or provide TEM EELS spatial maps showing F everywhere, or provide a discussion of what happens to the amorphous carbon.

A key challenge of a multilayer device composed of individual layers of 2D crystals is to make electrical contact to embedded layers through "vias." The authors show how their technique achieves conducting vias, and argue that the conduction path circumvents the non-conducting GF and is achieved through a G edge. Their measurements for different edge widths and DFT calculations confirm this hypothesis.

The authors go on to demonstrate several different applications of their etching technique, including an electrically contacted G channel embedded in hBN, a G-hBN-G "capacitor", a complex multichannel, multi-layer heterostructure device, and a GF mechanical resonator. Through using their technique, the authors achieve a record and near theoretically limited resistivity/mobility in the hBN embedded G device. The hBN embedded G device is an impressive application of the GEM technique. Although these devices appear impressive, they authors do not spend enough time motivated the need for any of them. They provide more quantity of applications than quality of describing them and their utility. Because of this, the paper loses focus and feels scattered. For example, the discussion of the complex multichannel, multi-layer heterostructure device (see Fig. 4a) would benefit from more motivation. An analogy to multilayer PCBs could be good, for example, and it would be nice if the authors provided a few concrete examples of why the structure may be useful. The same is true for the FG mechanical structure; it's a lot of work to make the mechanical structure and the benefits of the approach are not at all clear. Also, the authors do not provide a phase response curve to accompany the amplitude curve or the FG mechanical resonator. The correct phase response is needed to rule out spurious resonances.

The GEM technique is novel and it will be of interest to the field of 2D heterostructures and high-mobility G devices. Because of the way the paper is written, those interested in high-mobility G devices may not become aware of it; the high-mobility device becomes one of many reported applications of the technique. The work will also have a limited reach because the "big picture" is fuzzy; the first two sentences of the introductory paragraph are too abstract and thus lack concrete examples of where and why their GEM technique will be useful. The resolving paragraph is similarly vague. To expand their impact, the authors need to identify and clearly articulate concrete ways that their work will advance science or technology. The authors do provide sufficient details to reproduce the fabrication and measurements reported in their paper.

My biggest scientific criticism is that it's unclear if the GEM is necessary to make some of the structures reported in the paper. As part of this criticism, the authors provide a very limited background discussion (literally, one sentence) of previous and current methods to achieve the same result; this is a weakness of their paper. It would be nice if the authors could argue why it is better than straight lithography, and explicitly point out when straight lithography will fail. In fact, they are still using resist-based lithography for most of their devices. For example, the conducting via structures could be made without any GEM properties at all, and just use traditional resist-based lithography.

As an issue of clarity and helping to guide the reader, the authors may want to provide more motivation in paragraphs that begin with "Figure X shows, illustrates, etc." There are about 6 of these paragraphs at the beginning of the manuscript, and it takes some rereading to figure out where the story is going. As a reader, I had to try extra hard to follow the paper because key yet simple connecting pieces were missing.

Reviewer #3 (Remarks to the Author):

The manuscript entitled "Atomically-precise graphene etch masks for 3D integrated systems from 2D material heterostructures" is submitted to the Nature Communications journal. The paper is related to the field of fabrication of carbon-based and other 2D materials heterostructures. It will be of interest to a broad graphene research community. The paper is written in good English, is well illustrated, and a short introductory paragraph is written, all necessary details on the sample fabrication and measurements are provided, the conclusions are supported by the experimental evidence. The main result of this paper is the demonstration of the reliable procedure to fabricate polymer-free heterostructures based on graphene etch stop layer. The paper provides a significant breakthrough in fabrication technology. An interesting insight was found to be useful as an essential step for high-mobility devices and for MEMS structures fabrication. This paper deserves publication in the Nature Communications journal after minor correction.

It is known that the mobility in graphene almost independent of the concentration of the charged carriers. Then if no other mechanisms of carrier scattering such as remote impurities, phonons, resonant scatterers, point-like defects, which usually are discussed in the literature, we do not expect an increase of the mobility near the Dirac point. However, the analysis of the data in Figure 3 (lines 126-134) is based on the conductivity obtained from four-terminal measurements. I need to point out that the geometry is not well defined. It is not a standard Hall Bar which is used for Hall effect measurements. The contacts itself can dope the regions of the sample near the metal and reduce conductance, therefore, calculations of the conductivity near the Dirac point cannot be correct. The other reason why standard approach with a single type of carriers is not valid is the presence of electron-hole puddles which can produce $1e11 \text{ cm}^{-2}$ concentration variation. The most reliable data are shown in the Supplementary materials where low-temperature Hall Bar based measurements are presented. Therefore I suggest softening the statement about the mobility measured at room temperature to the region of high concentration where the data are more reliable.

Detailed response to the questions raised by the reviewers

Reviewer 1

General remarks of Reviewer 1:

The manuscript by Son et al. outlines a technical process through which graphene can serve as an etch mask for lithographically fabricating 2D material devices. XeF₂ gas is known to etch materials such as h-BN, TMDs, and phosphorene, while only fluorinating graphene to leave its carbon backbone intact such that underlayers are protected from etching. Several devices are fabricated and electrically tested to show how the process proceeds under different geometrical considerations. Electrically tested graphene devices show high mobility after processing.

Comment #1: Overall the paper is well written and the experiments appear to be thoughtfully executed. The central point of graphene serving as an etch mask to XeF₂ is in and of itself not a new finding, though this work extends the examples and complexity through which graphene can be used to form device structures, in particular focusing on multi-layer, metal-patterned structures. As such, the introduction must make this point clearer with a few sentence discussion/referencing of how graphene has been used in the past specifically as an etch mask with XeF₂. This work has the potential to further motivate researchers to implement graphene/XeF₂ etching techniques.

For example: (i) In Reference 5, Figure 4 (and Figure S4) they use graphene with O₂ plasma etched holes to XeF₂ etch a MoS₂ underlayer, which is analogous to the experiment shown in this manuscript in Figure 2; (ii) In Reference 22, Figure 2 they use graphene as an etch mask and SOI as a sacrificial underlayer, as well as patterning the fluorine itself on graphene using photoresist; (iii) In *Nano Lett.* **12**, 4212 (2012), Figure 2 they use reduced graphene oxide and in *ACS Nano* **11**, 4745 (2017), Figure 1 they use CVD graphene with lithographically etched holes to undercut a SOI sacrificial underlayer to form arrays of mechanical drum resonators, which is analogous to the experiment shown in this manuscript in Figure 4. These last references also study how chemistry impacts graphene's properties in mechanical resonators, which is relevant for graphene mechanical structures formed via XeF₂ etching.

Response to comment #1: We thank the reviewer for correctly pointing out that we should have made a more thorough background section in our paper. We have created a new paragraph which discusses relevant results with fluorination and etch masks from graphene, and explicitly explain how our process is different from these previous results. We had already cited some of the papers the reviewer suggested, but have made sure to include the two that we missed initially. In addition, to make it clearer what our innovation in the paper is, we have changed our paper title and process from graphene etch masks (which has been demonstrated in specific cases) to graphene etch *stops* which is novel.

We would like to point out how our process is novel. There have been several papers in which XeF₂ and graphene are used for lithography and etch process. In reference 5 and 8 in revised manuscript (reference 22 in initial manuscript), graphene was used as etch masks for etching MoS₂ and underlying silicon. In two papers from *Nano Lett.* **12**, 4212 (2012) and *ACS Nano* **11**, 4745 (2017), graphene and XeF₂ were used to fabricate suspended graphene drum structures by etching underlying SOI.

To our knowledge, graphene has never been used as an *etch stop* layer, and has never been used as a

fabrication technique for contacting buried layers in 2D heterostructures. Specifically, our results are distinct from previous demonstrations of graphene etch masks by enabling new fabrication techniques and new discoveries:

- (i) We used one-atom-thick graphene etch stop (or mask) layers to precisely pattern 2D heterostructures and electrically access buried layers with one-atom-thickness level precision.
- (ii) While fluorinated graphene is an insulator, we found the surprising result that metalized FG contacts lead to extremely low resistance contacts, down to 80 ohm· μm . We find that the strong overlap of the fluorine and metal orbitals acts as a bridge to lead to efficient transfer of electrons into the graphene. As a result, the buried FG-metallized contacts behave as one-dimensional edge contacts.
- (iii) Combining the fully encapsulated graphene and one dimensional contacts, the embedded device has performance as good as state of the art and approaching theoretical limits.
- (iv) We demonstrate that the etch stops enable new fabrication capabilities such as interlayer vias and separately contacting multiple active layers. This fabrication technique does not require carefully offsetting layers during heterostructure fabrication, which is the current method for contacting vertically offset layers. This is a key capability for scalably engineering vertically integrated electronics like integrated circuit logic (e.g. NAND gates) or out of plane transport (e.g. tunnel junctions or PN diodes). The lithography process for via contact fabrication is straightforward, clean, and robust leading to the extremely small contact resistance.
- (v) As pointed out by the reviewer, there have been a couple of papers which showed suspending graphene membranes using graphene etch masks on SOI. Here we extend this concept to show that it can be applied to 2D heterostructures as well. Here the interesting discovery is that phosphorene can be used as a sacrificial release layer which undercuts the graphene, a critical capability in MEMS/NEMS technologies.

We believe that graphene *etch stops* is a simple, and ultraclean new technique which will motivate many researchers within the field of 2D heterostructures.

As the reviewer suggested, we have added a new background paragraph highlight and added additional references on previous work on graphene etch masks.

[On page 4, line 5th in Manuscript] “This new concept takes advantage of the high chemical selectivity of XeF₂, a vapor phase, strong fluorinating agent commonly used as an isotropic etchant for silicon in the microelectromechanical systems industry²¹. Several 2D materials including hBN and transition metal dichalcogenides (TMDs), are quickly etched when exposed to XeF₂^{16,22,23}. In contrast, graphene reacts with XeF₂ to form fluorographene (FG)²⁴⁻²⁷, a wide-band-gap semiconducting monolayer²⁸ with composition C₄F, in the case that only one side is exposed²⁴. There have been several demonstrations which take advantage of this selectivity to use graphene as an etch mask for shaping MoS₂¹⁶, as a mask to etch underlying silicon²⁹⁻³¹, and to create a sacrificial release layer to suspend graphene drumhead resonators on SOI¹⁰. Our innovation has been to apply this etch selectivity to access buried graphene layers embedded within the heterostructures and as masks for patterning the underlying layers.”

Comment #2: Another point not explicitly discussed relates to the stability of graphene fluorinated by XeF₂ [Reference 8]. It would be helpful to the reader to note that the fluorination is somewhat unstable and will change over time in ambient, as well as to subsequent processing in lithography-based solvents or vacuum. If the reader expects to have a ‘locked-in’ graphene fluorination level after XeF₂ exposure and processing, that will likely not happen.

Response to comment #2: We agree with the reviewer that the stability of fluorinated graphene is an important consideration and have added a supplementary figure S8, reproduced below (Figure R1) which explicitly addresses the stability of our samples. Previous work has shown that partially fluorinated graphene is unstable, but fully fluorinated graphene is relatively stable. (*Nano Lett.* **13**, 4311-4316 (2013)). Our results are consistent with the observations in this paper. We observe that the stability of fluorinated graphene that is exposed to air depends strongly on the exposure time. Under longer exposures of 720 s, we observe that fluorinated graphene maintains its Raman signature and insulating properties for more than two months in ambient conditions. However, it should be noted that partially fluorinated graphene (treated for 5 s) experiences a structural change in ambient condition, probably due to formation of FG islands and localized stress.

Moreover, in our original manuscript (Fig. S11 originally, now Fig. S12), we observed that the embedded fluorinated graphene contacts maintained the low contact resistance after one month in ambient conditions.

Figure R1. The stability of fluorographene (FG). The Raman spectra of (a) slightly and (b) highly fluorinated graphene treated by XeF₂ gas for 5 s and 720 s, respectively. The highly fluorinated graphene is stable up to 2 months in ambient condition, while the slightly fluorinated graphene is unstable. (c) Highly fluorinated graphene maintains its high resistance over 2 months in ambient condition.

Following the reviewer's suggestion, we modified the manuscript regarding the stability of FG and added the Supplementary Figure (Fig. S8).

[On page 11, in Supplementary Information] We added Fig. R1 as the Supplementary Figure (Fig. S8).

[On page 7, line 1st in Manuscript] We added “As shown in Supplementary Fig. S8, when graphene on hBN is functionalized for longer periods (720 s), it maintained its structure and high resistance for over two months in ambient conditions.”

Reviewer 2

General remarks of Reviewer 2:

The authors present a technique to selectively etch and pattern hexagonal boron nitride (hBN) and tungsten disulfide (WS_2) by using xenon difluoride (XeF_2) gas and a graphene etch mask (GEM). They claim that graphene is fluorinated but not etched by the XeF_2 , and acts as an impermeable barrier to the XeF_2 gas. Therefore, the XeF_2 etches the hBN or WS_2 that is not protected or masked by the GEM. In any case, they use the GEM to deterministically etch hBN and WS_2 . The authors argue that the GEM technique addresses a need to address individual layers of a 2D crystal heterostructure.

Comment #1: To show that the graphene becomes fluorinated, the authors perform Raman and XPS, and they also fabricate a G/hBN FET. Raman and XPS clearly show fluorination directly or indirectly, and the conduction of the FET falls sharply with increased XeF_2 exposure. To show that G isn't etched, they perform TEM and also show optical images of the devices. From both, it is clear that G is not etched. They also show TEM cross-sections of etched hBN and WS_2 , and demonstrate nanometer-scale control and a very small undercut. The authors also claim that the fluorination of graphene is continuous, which they attempt to support with TEM; I'm not convinced by this claim, and wonder what happens to the amorphous carbon (PMMA residue) on the graphene. Does it become fluorinated and thus protect the graphene below it? Perhaps the authors could soften the claims of uniformity, or provide TEM EELS spatial maps showing F everywhere, or provide a discussion of what happens to the amorphous carbon.

Response to comment #1: The reviewer is correct that the TEM image does not confirm the uniformity or continuity of fluorination, and have softened these claims in the papers. Generally, we agree with the reviewer that the fluorographene is not perfectly continuous or uniform on the nanoscale. For example, previous reports have shown fluorination clusters on graphene resolved by conductive AFM by K. S. Novoselov (springer, 2016)). As with any experiments with chemical functionalization of 2D materials, residue from transfer or lithography will upset the uniformity. Moreover, there is more than one stable phase of fluorinated graphene and the fluorination process leads to crystal frustration of the different phases. We do note that the fluorination is fairly uniform on the microscale, as shown by Raman maps in Figure 1b inset.

We have left the TEM image of the sample in Figure S6, because it serves another important message. It confirms that the XeF_2 gas treatment does not lead to the formation of voids or pores, which supports our claim that the etch stop is impermeable. This is why we do not observe damage in the hBN underneath the etch stops, as shown in Figure 1d and e.

Figure R2. Raman signals of graphene and Fluorographene (FG) on TEM grid.

As the reviewer mentioned, in any 2D materials sample created for TEM, there is frequently amorphous carbon residue (typically PMMA) leftover from transfer. To minimize this residue, we annealed the transferred graphene sample before exposure to XeF₂. There is still a small amount of PMMA residue, but we do not observe any voids or pores in the polymer free regions.

We also tried TEM EELS analysis on our samples as the reviewer suggested. But, because of the defluorination of graphene under the high energy of the electron beam, it was difficult to get clear data on the fluorine distribution.

To address the reviewers concern, we have softened our claims of uniformity or continuity with the following specific changes to the text:

[On page 4, line 9th in Manuscript] We removed the word, ‘continuous’, in the sentence.; “In contrast, graphene reacts with XeF₂ to form fluorographene (FG)²⁴⁻²⁷, a wide-band-gap semiconducting monolayer²⁸ with composition C₄F, in the case that only one side is exposed²⁴.”

[On page 9, caption of Fig. S6 in Supplementary Information] We changed the caption of Fig. S6; “**Fig. S6 | HR-TEM image of FG.** The HR-TEM image shows crystal lattice of CVD graphene fluorinated by exposure to XeF₂ for 100 s. The CVD graphene was transferred onto a TEM grid with PMMA transfer method, followed by annealing (340°C, 4h) to remove PMMA residue. However, small amounts of PMMA residue are still observed as indicated by arrows. After exposure to XeF₂ gas, no voids were observed in the fluorinated region. The Fast Fourier Transform (FFT) in the inset shows that fluorinated graphene maintains crystallinity after fluorination. This confirms that fluorination process makes no voids in FG and FG can be used as an impermeable membrane to the XeF₂.”

»

Comment #2: The authors go on to demonstrate several different applications of their etching technique, including an electrically contacted G channel embedded in hBN, a G-hBN-G "capacitor", a complex multichannel, multi-layer heterostructure device, and a GF mechanical resonator. Through using their technique, the authors achieve a record and near theoretically limited resistivity/mobility in the hBN embedded G device. **The hBN embedded G device is an impressive application of the GEM technique.** Although these devices appear impressive, they authors do not spend enough time motivated the need for any of them. They provide more quantity of applications than quality of describing them and their utility. Because of this, the paper loses focus and feels scattered. For example, the discussion of the complex multichannel, multi-layer heterostructure device (see Fig. 4a) would benefit from more motivation. An analogy to multilayer PCBs could be good, for example, and it would be nice if the authors provided a few concrete examples of why the structure may be useful.

Response to comment #2: We appreciate the feedback from the reviewer that the manuscript became defocused with so many device demonstrations discussed, and have worked to add clarity and context to improve the readability. In order to clarify the common motivations for the different demonstrations, we have changed the manuscript introduction to focus on the utility of etch stops in integrated circuitry and out of plane integration, and added transition paragraphs before each demonstration to make clear how they are related and concrete examples of how they may be used. We enumerate the changes below:

[On page 2, line 17th in Manuscript] We added "As next generation technologies of electronic, photonic, and mechanical devices approach the atomic scale, it is important to develop atomically-precise fabrication methods. Among them, etch stops, critical for the vertical integration of nanoelectronic and nanomechanical devices, are created by layering materials with drastically different etch properties and embedded into a structure, allowing for patterning feature sizes, accessing buried layers, or undercutting to create suspended structures."

[On page 3, line 2nd in Manuscript] We added "Many of the applications for 2D material heterostructure devices demand out of plane integration^{5,6}, contacting multiple active layers, and creating interconnects between the different layers. Examples include 2D material based integrated circuitry like NAND gates⁴ or ring resonators⁷; devices based on interlayer tunneling like light emitting diodes⁸ or tunnel junctions⁹; and nanoelectromechanical systems like drumhead resonators based on atomic membranes¹⁰."

[On page 7, line 5th in Manuscript] We added "Two persistent challenges in nanoscale device research are how to minimize the impact of environment on limiting the potentially outstanding electronic mobility of nanomaterials, and how to engineer low resistance contacts to nanomaterials."

[On page 7, line 12th in Manuscript] We added "In Fig. 3a, we demonstrate the application of GES to electrically contact a buried graphene layer encapsulated in hBN."

[On page 10, line 8th in Manuscript] We added "In addition to offering a simpler fabrication process and state of the art device properties, the selective etch stop also enables new capabilities which cannot be easily or scalably realized using other techniques. For example, interlayer vias and independently contacting multiple active layers in vertically aligned heterostructures are critical to integrated circuits like NAND gates⁴, where logic operations are computed by coupling the gates and channels of several transistors in series, and graphene-based multilayered printed circuit boards (PCBs). Similarly, many device applications of 2D heterostructures which rely out of plane transport,

like PN diodes³⁸, tunnel junctions⁹, or light emitters based on 2D materials^{8,39} require the same ability to contact vertically aligned layers separately.”

[On page 11, line 7th in Manuscript] We added “The contact resistance is negligibly small compared with the corresponding channel resistances, demonstrating that GES enables efficient, simple and selective contacts to vertically offset layers to create low resistance interlayer vias which could be integrated to the complicated devices such as multilayer printed circuit boards (PCBs) or light emitter based on 2D materials²⁹.”

[On page 13, line 17th in Manuscript] We added “All of these components suggest that the selective etch process is a major capability necessary for the realization of atomically-precise, all-2D nanoelectronics as a viable technology, in applications where vertical integration is critical, like integrated circuit logic components (e.g. NAND gates), devices operating through out of plane transport (e.g 2D material tunnel junctions and LEDs), and in 2D nanoelectromechanical systems.”

Comment #3: The same is true for the FG mechanical structure; it's a lot of work to make the mechanical structure and the benefits of the approach are not at all clear. Also, the authors do not provide a phase response curve to accompany the amplitude curve or the FG mechanical resonator. The correct phase response is needed to rule out spurious resonances.

Response to comment #3: Here we want to show that our approach can give a straightforward way toward fabrication of suspended graphene structure, one application of which is a resonator. The advantage of this fabrication process is that it is all-dry process and it can be used for integration of suspended graphene structures for analog devices. One of the most common applications of XeF₂ in industry is as a selective etchant for suspending MEMS structures, so we felt it would be remiss to not demonstrate the all-2D version of this application in our paper.

Regarding the phase response, we do not capture phase data because we use a spectrum analyzer in our measurements. However, these devices are measured using an optical interferometry detection technique which allows a superior check for whether the resonance is real that is not possible with pure electrical interrogation techniques where electrical resonance can occur in the measurement circuitry. Shown in Figure R3, by moving the laser on and off the resonator, we observe that the resonance will appear and disappear, which means that the resonance must be coming from the mechanical structure.

Figure R3. The measured dynamic reflected laser response depending on whether the laser is focused on the graphene membrane or on the substrate. Because the resonance only appears when on the suspended region, it must be a mechanical resonance and not a spurious electrical resonance.

We have made the following changes to the manuscript:

[On page 26 in Supplementary Information] We added Figure R3 as a new supplementary Fig. S21.

[On page 12, line 9th in Manuscript] We added “As a final demonstration of a different kind of vertical integration, a common application of selective etches in microelectromechanical systems is to suspend mechanically responsive structures. In studies on fluorination of graphene, it was shown that the graphene can be used as an etch mask for underlying silicon to generate suspended FG membranes²⁹⁻³¹. These atomic membranes behave as tensioned mechanical drumhead resonators, useful as low mass chemical sensors or tunable radio frequency filters or oscillators⁴¹. In Fig. 4e, we show that the same concept can be applied to fabricate suspended graphene NEMS from 2D heterostructures.”

Comment #4: The GEM technique is novel and it will be of interest to the field of 2D heterostructures and high-mobility G devices. Because of the way the paper is written, those interested in high-mobility G devices may not become aware of it; the high-mobility device becomes one of many reported applications of the technique. The work will also have a limited reach because the "big picture" is fuzzy; the first two sentences of the introductory paragraph are too abstract and thus lack concrete examples of where and why their GEM technique will be useful. The resolving paragraph is similarly vague. To expand their impact, the authors need to identify and clearly articulate concrete ways that their work will advance science or technology. The authors do provide sufficient details to reproduce the fabrication and measurements reported in their paper.

My biggest scientific criticism is that it's unclear if the GEM is necessary to make some of the structures reported in the paper. As part of this criticism, the authors provide a very limited background discussion (literally, one sentence) of previous and current methods to achieve the same result; this is a weakness of their paper. It would be nice if the authors could argue why it is better than straight lithography, and explicitly point out when straight lithography will fail. In fact, they are still using resist-based lithography for most of their devices. For example, the conducting via structures could be made without any GEM properties at all, and just use traditional resist-based

lithography.

[Response to comment #4] We thank the reviewer for the great advice on how to make the paper clearer. The two comments on articulating the ways in which the etch stops advances science and technology, and where the etch stops is necessary are related so we will address them together. We agree with the reviewer that some of the applications of the etch stop technique are subtle unless you have to fabricate 2D heterostructures and have to regularly deal with the many practical considerations needed to produce quality functional devices. We have significantly re-written the introduction to directly relate the typical applications of etch stops as a motivation which defines and relates the different device demonstrations shown, and added contextualizing paragraphs between the different device demonstrations. In addition, we have added additional background on the state of the art methods to contact 2D materials.

Now we will articulate how GES is necessary to make the structures reported in the paper.

First, to clarify the discussion, the GES technique is not a replacement for lithography, it is a replacement for etch methods and for the planning and offsets that must be put into stacking heterostructures before lithography. For nearly any device application, lithography must be used in conjunction with etching and metal evaporation to produce devices.

Background:

The challenge in fabricating devices in 2D heterostructures with more than one active layer is how to contact each active layer separately, without shorting to any of the intermediate layers. The majority of device demonstrations use exfoliated materials where each layer is carefully offset so that they stick out from underneath the layers on top somewhere in the heterostructure, then lithography is used to separately contact each layer. While this technique works for single demonstrations of device concepts, it is extremely laborious and not scalable to more than one device at a time because each layer must be separately exfoliated and chosen for size and shape to ensure the offsets. Moreover, to get the highest mobility out of most 2D materials, it is necessary to completely encapsulate the material in insulating hBN so that the material never sees solvents or polymers, so the offset techniques do not work. In fact, many emergent 2D materials (e.g. Phosphorene, NbSe₂, Cr₂I₃) are air sensitive and must be fully encapsulated with hBN while inside a glove box before fabrication. The current state of the art method is to use “edge contacts” where heterostructures are etched through to expose the edges of buried layers, then metals are evaporated onto the sides of the heterostructure to make one-dimensional contacts [*Science* **342**, 614-617 (2013), *APL Mater.* **2**, 1056105 (2014)]. This method has led to a dramatic improvement in the mobility and quality of electronic devices because it allows contact to electronic layers which are fully encapsulated and thus have a minimum of disorder.

Strength of GES compared with edge contacting:

The relative strength of GES versus edge contacting is graphically illustrated in Figure R4. Edge contacts still require careful offsetting of each active layer because the etching is not selective so all vertically aligned layers in the heterostructure are exposed simultaneously. In contrast, unlike the non-selective edge contact method, the GES technique allows selective and separate contact to vertically offset layers because it will stop on the topmost graphene layer.

Figure R4. The comparison of the edge contacting and GES.

The interlayer vias and contacting of multiple vertically offset layers are demonstrations of how the etch stops may be applied to engineer devices. They should be thought of as demonstrations of basic capabilities which could be applied to a wide variety of different applications. It is, of course, possible to make interlayer vias or to separately contact multiple layers if one starts with the right heterostructure with the layers appropriately offset. The key here is that is no longer necessary with the etch stops.

To enumerate the key discoveries of this paper, and how they push forward science and technology,

To our knowledge, graphene has never been used as an etch stop layer, and has never been used as a fabrication technique for contacting buried layers in 2D heterostructures. Specifically, our results are distinct from previous demonstrations of graphene etch masks by enabling new fabrication techniques and new discoveries:

- (i)** We used one-atom-thick graphene etch stop (or mask) layers to precisely pattern 2D heterostructures and electrically access buried layers with one-atom-thickness level precision.
- (ii)** While fluorinated graphene is an insulator, we found the surprising result that metalized FG contacts lead to extremely low resistance contacts, down to 80 ohm-um. We find that the strong overlap of the fluorine and metal orbitals acts as a bridge to lead to efficient transfer of electrons into the graphene. As a result, the buried FG-metallized contacts behave as one-dimensional edge contacts at the 1D edge interface between the graphene-fluorographene rather than surface contacts at the 2D metal-graphene interface. Therefore the contact resistance should only depend on channel width rather than metal area which should be beneficial to scaling down of devices to nm dimensions.
- (iii)** Combining the fully encapsulated graphene and one dimensional contacts, the embedded devices have performance as good as state of the art and approaching theoretical limits.
- (iv)** We have demonstrated basic fabrication capabilities for which etch stops are frequently used in silicon and other nanotechnologies as methods to achieve vertical integration of devices: Contacting embedded or buried layers, interlayer vias, separately contacting vertically offset active layers without shorting, and undercutting to suspend nanostructures with a sacrificial layer.
- (v)** This fabrication technique does not require carefully offsetting layers in plane during heterostructure fabrication, which is the current method for contacting vertically overlapping layers. This is a key capability for scalably engineering vertically integrated electronics like integrated circuit logic (e.g. NAND gates) or out of plane transport (e.g. tunnel junctions or PN diodes). The lithography process for via contact fabrication is straightforward, clean, and robust leading to the extremely small contact resistance.
- (vi)** Because of the simplicity of this process, it represents a method for producing vertically integrated devices from 2D heterostructures which is simpler, more reliable, and more scalable than current state of the art methods. We demonstrate this through a proof of concept patterning of large area MoS₂.

We believe that graphene etch stops is a simple, and ultraclean new technique which will motivate many researchers within the field of 2D heterostructures.

In addition to the changes already discussed in Comment 2, we have made the specific changes to our manuscript to address the reviewer's comments.

[On Page 21 in Supplementary Information] We added Fig. R4 as Fig. S17 in the Supplementary Information to help the readers fully understand our technique.

[On Page 3, line 14th in Manuscript] We added more sentences in the manuscript as follows. "The current state of the art method is to use "edge contacts" where heterostructures are etched through to expose the edges of buried layers of graphene encapsulated in insulating hexagonal boron nitride (hBN), then metals are evaporated onto the edge to make one-dimensional contacts¹¹. This method has lead to a dramatic improvement in the mobility and quality of electronic devices because it allows contact to electronic layers which are fully encapsulated and thus have a minimum of disorder^{11,15}. However, edge contacts still require careful offsetting of active layer because the etching is not selective so all vertically aligned layers in the heterostructure are exposed simultaneously. A method which combines the superior device behavior of the edge contacts, but which simultaneously allows ready patterning of 2D heterostructures from large area continuous sheets and individually addressing of each layer are critical for translating many of the recent demonstrations of this new class of devices into scalable technologies."

[On Page 4, line 19th in Manuscript] We added more sentences in the manuscript as follows. "This new capability enables simple and scalable methods to vertically integrate 2D devices through contacting multiple active layers, interlayer vias and suspended nanostructures, yet maintains the state of the art performance of fully encapsulated 2D devices."

[On Page 10, line 3rd in Manuscript] We added a sentence to emphasize importance of FG via contact for scaling down. "Unlike in two dimensional surface contacts, this result indicates that the contact resistance of the metalized FG is independent of the contact length and there is no intrinsic lower limit of size. Hence it should be possible to scale down the size of embedded contacts and vias to nanoscale dimensions without impacting functionality."

Comment #5: As an issue of clarity and helping to guide the reader, the authors may want to provide more motivation in paragraphs that begin with "Figure X shows, illustrates, etc." There are about 6 of these paragraphs at the beginning of the manuscript, and it takes some rereading to figure out where the story is going. As a reader, I had to try extra hard to follow the paper because key yet simple connecting pieces were missing.

Response to comment #5: We thank the reviewer for this comment. This helps us to improve readability of our manuscript. To help the readers follow the story, we revised the manuscript by providing contextualizing sentences and paragraphs between each measurement as the reviewer suggested.

[On page 5, line 11th in Manuscript] We revised the sentence as follows. "To examine the selectivity and resolution limits of GES, we obtained cross-sectional images of the etched heterostructures with a scanning transmission electron microscope (STEM), as shown in Fig. 1d and 1e."

[On page 5, line 22th in Manuscript] We revised the sentence as follows. "The self-arresting nature of GES means that it is scalable as well as being atomically-precise. Figure 2a demonstrates this

scalability by applying GES to a large area heterostructure array. We patterned large area graphene as etch masks for patterning large area WS₂, both grown by chemical vapor deposition (CVD).”

[On page 6, line 10th in Manuscript] We revised the sentence as follows. “Before examining the application of GES to 2D heterostructure devices, we confirm the structure and electrical properties of FG.”

[On page 7, line 4th in Manuscript] We revised the sentence as follows. “In the rest of the paper, we will explore the application of the graphene etch stops to fabricating electronic and mechanical devices from 2D heterostructures. First, in Fig. 3, we demonstrate the application of GES to electrically contact a buried graphene layer encapsulated in hBN layers. Two persistent challenges in nanoscale device research are how to minimize the impact of environment on limiting the potentially outstanding electronic mobility of nanomaterials, and how to engineer low resistance contacts to nanomaterials.”

[On page 10, line 17th in Manuscript] We revised the sentence as follows. “Figure 4 outlines proof-of-concept demonstrations of using GES to fabricate interlayer vias and vertically integrate multiple active layers.”

[On page 11, line 12th in Manuscript] We revised the sentence as follows. “The second demonstration takes advantage of the combined high in plane resistance of FG with the low contact resistance when the FG is metalized. Through sequential patterning and etching steps, GES allows independent contact of multiple active layers which interact to generate device functionality.”

Reviewer 3

General remarks of Reviewer 3:

The manuscript entitled “Atomically-precise graphene etch masks for 3D integrated systems from 2D material heterostructures” is submitted to the Nature Communications journal. The paper is related to the field of fabrication of carbon-based and other 2D materials heterostructures. It will be of interest to a broad graphene research community. The paper is written in good English, is well illustrated, and a short introductory paragraph is written, all necessary details on the sample fabrication and measurements are provided, the conclusions are supported by the experimental evidence. The main result of this paper is the demonstration of the reliable procedure to fabricate polymer-free heterostructures based on graphene etch stop layer. The paper provides a significant breakthrough in fabrication technology. An interesting insight was found to be useful as an essential step for high mobility devices and for MEMS structures fabrication. This paper deserves publication in the Nature Communications journal after minor correction.

Comment #1: It is known that the mobility in graphene almost independent of the concentration of the charged carriers. Then if no other mechanisms of carrier scattering such as remote impurities, phonons, resonant scatterers, point-like defects, which usually are discussed in the literature, we do not expect an increase of the mobility near the Dirac point. However, the analysis of the data in Figure 3 (lines 126-134) is based on the conductivity obtained from four-terminal measurements. I need to point out that the geometry is not well defined. It is not a standard Hall Bar which is used for Hall effect measurements. The contacts itself can dope the regions of the sample near the metal and reduce conductance, therefore, calculations of the conductivity near the Dirac point cannot be correct. The other reason why standard approach with a single type of carriers is not valid is the presence of electron-hole puddles which can produce $1 \times 10^{11} \text{ cm}^{-2}$ concentration variation. The most reliable data are shown in the Supplementary materials where low-temperature Hall Bar based measurements is presented. Therefore I suggest softening the statement about the mobility measured at room temperature to the region of high concentration where the data are more reliable.

Response to comment #1: We thank the reviewer for their high evaluation of our work. We agree with the reviewers point about our device geometry and our mobility estimation at low carrier concentration. To address this comment, we have replaced the device measurements in Figure 3 on the 4-point geometry with measurements on Hall bar geometry, and have softened our statements on mobility to only cite the high-carrier density mobilities at room temperature.

We removed the mobility of $140,000 \text{ cm}^2\text{V}^{-1}\text{s}^{-1}$ at low concentration and room temperature, and changed our statements as below.

[On page 27, Fig. 3b in Manuscript] We replaced previous figure with new data measured with Hall bar geometric device.

[On page 2, line 10th in Manuscript] We modified the sentence as “Embedded graphene transistors show a room temperature mobility of $\sim 40,000 \text{ cm}^2\text{V}^{-1}\text{s}^{-1}$ at carrier density of $n \sim 4 \times 10^{12} \text{ cm}^{-2}$ and low contact resistivity of $80 \text{ } \Omega \cdot \mu\text{m}$, both approaching theoretical limits.”

[On page 4, line 16th in Manuscript] We added the sentence as “Surprisingly, the embedded contacts, which is composed of FG-metal contacts, lead to room temperature carrier mobilities of $\sim 40,000$

$\text{cm}^2\text{V}^{-1}\text{s}^{-1}$ at carrier density $n \sim 4.0 \times 10^{12} \text{ cm}^{-2}$, and behave as one dimensional contacts with low contact resistivity of $80 \text{ } \Omega \cdot \square\text{m}$, approaching theoretical limits^{11,32,33}. This new capability enables simple and scalable methods to vertically integrate 2D devices through contacting multiple active layers, interlayer vias and suspended nanostructures, yet maintains the state of the art performance of fully encapsulated 2D devices.”

[On page 7, line 23th in Manuscript] We revised the sentence as “Figure 3b shows the field-effect characteristics of a graphene Hall bar device encapsulated by hBN with FG contacts. At high carrier concentration of $n \sim 4.0 \times 10^{12} \text{ cm}^{-2}$, the sheet resistance was $\sim 45 \text{ } \Omega/\square$ at room temperature, corresponding to a carrier mobility of $\sim 40,000 \text{ cm}^2\text{V}^{-1}\text{s}^{-1}$, close to the theoretical limit^{32,33}. As shown in the inset of Fig. 3b, the mobility drastically increases with decreasing carrier concentrations, as expected from the acoustic-phonon-limited model^{11,32,33}. On a Hall bar device measured at low temperature $T=1.7 \text{ K}$ (Fig. S10), the low carrier concentration mobility increased to $\sim 460,000 \text{ cm}^2\text{V}^{-1}\text{s}^{-1}$.”

- End of responses -

REVIEWERS' COMMENTS:

Reviewer #1 (Remarks to the Author):

Overall the authors appear to have addressed technical questions/comments from all Reviewers adequately. However, there are instances in the text where references are applied, but the referenced paper is not directly related to the written claim. It gives the impression that the authors lack attention to detail by applying references for work that is tangentially related (i.e. merely "close enough"), and/or that the authors do not know the original work in the field. For example:

Line 83-84: "...and to create a sacrificial release layer to suspend graphene drumhead resonators on SOI [ref 10]." Reference 10 does not contain information for using SOI as a sacrificial release layer.

Line 282: "...can be used as an etch mask for underlying silicon to generate suspended FG membranes [ref 29-31]." References 29-31 do not contain information on forming suspended FG membranes.

The authors need to improve how they reference prior work in their manuscript. The lack of attention to detail is noticeable and reduces confidence that there are not other errors with the referencing. It would be premature to accept this manuscript in Nature Communications without improving how references are applied.

Reviewer #2 (Remarks to the Author):

The authors have done an excellent job of responding to all of my original comments. I recommend publication.

Reviewer #3 (Remarks to the Author):

All my comment were taken into account into the new version of the paper and I do not see any new problems. I suggest to publish this paper as it is.

Detailed response to the questions raised by the reviewers

Reviewer 1

General remarks of Reviewer 1:

Overall the authors appear to have addressed technical questions/comments from all Reviewers adequately. However, there are instances in the text where references are applied, but the referenced paper is not directly related to the written claim. It gives the impression that the authors lack attention to detail by applying references for work that is tangentially related (i.e. merely “close enough”), and/or that the authors do not know the original work in the field.

The authors need to improve how they reference prior work in their manuscript. The lack of attention to detail is noticeable and reduces confidence that there are no other errors with the referencing. It would be premature to accept this manuscript in Nature Communications without improving how references are applied.

General response: The reviewer is completely correct in this comment, and we are grateful that they pointed out this unintentional oversight. In the 2nd revision, we combed through our references, changed the references that the reviewer pointed out, and found a few more instances of incorrect or inadequate references. The changes we have made are addressed below;

1. **Comment #1:** [Line 83-84] “..and to create a sacrificial release layer to suspend graphene drumhead resonators on SOI [ref 10].” Reference 10 does not contain information for using SOI as a sacrificial release layer.

Response to comment #1: We agree that this was not the appropriate reference. Accordingly, we’ve removed reference 10 and have referenced two other papers [reference 17: *ACS Nano* **11**, 4745-4752 (2017) & reference 22: *Nano Lett.* **10**, 3001-3005 (2010)]. These references were cited elsewhere in our manuscript but should have been cited here.

- Reference 17 [*ACS Nano* **11**, 4745-4752 (2017)]: In this paper, they used graphene nanomechanical resonators to investigate the mechanical properties of multilayer graphene films. To form graphene resonators, CVD graphene was transferred onto SOI substrates. Irrigation holes (1–5 μm) were etched through the graphene films to the top Si layer of the underlying SOI substrate, and the sample was then exposed to xenon difluoride (XeF_2) gas to undercut the silicon support layer and form suspended graphene resonators.
- Reference 22 [*Nano Lett.* **10**, 3001-3005 (2010)]: This work demonstrates the properties of fluorinated graphene. To provide the properties of double-side fluorinated graphene, they used graphene on SOI structure and fabricated suspended double-side fluorinated graphene after etching of underlying Si by XeF_2 gas.

[On page 4, line 13th in Manuscript] “...and to create a sacrificial release layer to suspend graphene membranes on SOI^{17,22}.”

2. **Comment #2:** [Line 282] “..can be used as an etch mask for underlying silicon to generate suspended FG membranes [ref 29-31].” References 29-31 do not contain information on forming suspended FG membranes.

Response to comment #2: We agree that these were not the appropriate references. Accordingly, we have removed references 29-31 and included the two other papers above [reference 17: *ACS Nano* **11**, 4745-4752 (2017) & reference 22: *Nano Lett.* **10**, 3001-3005 (2010)] at the end of the sentence.

[On page 13, line 3rd in Manuscript] “...can be used as an etch mask for underlying silicon to generate suspended graphene membranes^{17, 22}.”

3. **Other modifications to references:** Following the main comment above, we’ve went to over all other references to make sure each was adequate for the cited material. We made the following changes to the references.

- Two references (reference 1 (*Nature* **438**, 197-200 (2005)) and reference 2 (*Nat. Nanotechnol.* **6**, 147-150 (2011) in previous Manuscript) were removed as those do not specifically address 2D heterostructures but rather are applications of monolayer materials.
- Reference 7 [ACS Nano 7, 5588-5594 (2012)] in present Manuscript: We changed “ring resonator” to “ring oscillator” and changed the reference in the manuscript to a paper on 2D material based integrated circuitry.

[On page 3, line 4th in Manuscript] “...or ring oscillator⁷; devices based on interlayer...”

- Two references relating to the impermeability of graphene (reference 19 (*Nano Lett.* **8**, 2458-2462 (2008)) and reference 20 (*Carbon* **62**, 1-10 (2013)) in previous Manuscript) were moved to a more appropriate position and the reference number was changed (reference 30 and reference 31 in present Manuscript). References 24-27 in previous manuscript were removed from the sentence because those are not related to the impermeability of graphene after chemical modification. Instead, we added other paper (*Nano Res.* **6**, 200-207 (2013)) as reference 16 in present Manuscript.

[On page 5, line 22th in Manuscript] “These results demonstrate that FG maintains the impermeable nature of graphene³⁰⁻³¹ through the chemical modification process¹⁶.”

- All other reference numbers have been modified as appropriate.

Reviewer 2

General remarks of Reviewer 2:

The authors have done an excellent job of responding to all of my original comments. I recommend publication.

Reviewer 3

General remarks of Reviewer 3:

All my comments were taken into account into the new version of the paper and I do not see any new problems. I suggest publishing this paper as it is.

- End of responses -